materials science/nanotechnology/biomaterials

polylactic acid, graphene oxide, carboxylation, polyethylene glycol, mechanical properties

**Authors for correspondence:**
Huige Yang
e-mail: yanghg@zzu.edu.cn
Zhihao Chen
e-mail: chenzh@ztri.com.cn
Jinzhou Chen
e-mail: cjz@zzu.edu.cn

This article has been edited by the Royal Society of Chemistry, including the commissioning, peer review process and editorial aspects up to the point of acceptance.

# Polyethylene glycol grafted with carboxylated graphene oxide as a novel interface modifier for polylactic acid/graphene nanocomposites

Mingjun Niu[1], Hao Wang[1], Jing Li[1], Hongyan Chen[1], Lin Li[1], Huige Yang[1], Xuying Liu[1], Zhihao Chen[2], Hongzhi Liu[3] and Jinzhou Chen[1]

[1]School of Materials Science and Engineering, Zhengzhou University, Zhengzhou 450001, People's Republic of China
[2]Zhengzhou Tobacco Research Institute of CNTC, Zhengzhou 450001, Henan, People's Republic of China
[3]School of Chemical and Biological Engineering, NingboTech University, No. 1 Xuefu Road, Ningbo 315100, People's Republic of China

HY, 0000-0001-8004-6401; ZC, 0000-0002-1323-4645; JC, 0000-0002-2250-6921

Strength and toughness are both of great importance for the application of polylactic acid (PLA). Unfortunately, these two properties are often contradictory. In this work, an effective and practical strategy is proposed by using carboxylated graphene oxide (GC) grafted with polyethylene glycol (PEG), i.e. GC-g-PEG. The synthesis procedure of GC-g-PEG is firstly optimized. Then, a series of PLA nanocomposites were prepared by the melt blending method via masterbatch. In comparison to that achieved over pure PLA, these nanocomposites are of higher crystallinity, thermal stability and mechanical strength. This is mainly attributed to well-tailored interface and good dispersion. Especially, while retaining the tensile strength of the original PLA, the elongation at break increases by seven times by adding 0.3 wt% GC-g-PEG.

## 1. Introduction

Owing to the growing environmental concerns and feedstock pollution caused by petroleum-based non-degradable polymers, increasing attention has been drawn to the development of bio-based polymers as alternatives derived from renewable resources [1]. In view of sustainability, biocompatibility and biodegradability, polylactic acid (PLA) is identified as the most

promising alternative, which is a kind of degradable polyester, produced from raw agricultural materials such as flour and sugar [2]. Unfortunately, PLA, because of its intrinsic brittleness and low-tensile elongation at break, the application of PLA is still limited in spite of its high elastic modulus and tensile strength [3]. To address this issue, plenty of toughening strategies have been developed, including plasticization [4–6], copolymerization [7], blending with flexible polymers [8,9] and adding nanofillers [10,11]. Among them, nanofillers are widely used to improve the properties of PLA, such as strength and toughness [12]. This can be rationalized in terms of their nanosize, large specific surface area, high surface energy and functionality [13].

Graphene, as one of the most attractive nanofillers, is composed of a single-atom-thick graphite sheet, which has gained more and more attention owing to its superior mechanical strength, giant specific surface area, and distinguished electronic and gas barrier properties [14,15]. It is therefore endowed as a promising functional nanofiller to enhance the properties of polymers. However, uniform dispersion and effective interfacial interaction between the nanofiller and matrix are needed for obtaining ideal material properties [15–17]. Therefore, to achieve this purpose, plenty of efforts have been made on improving dispersibility and compatibility of graphene oxide (GO) in polymers [18,19]. Generally, the surface modification of GO is applied to adjust its surface properties and phase compatibility with polymers. Li *et al.* reinforced PLA with GO grafted with PLA (GO-g-PLA) via solution blending and then compression moulding. He found that, in comparison to the neat PLA, the elongation at break and tensile strength of PLA/GO-g-PLA nanocomposites was increased by 114.3% and 105.7%, respectively [20]. To date, the PLA/GO nanocomposites prepared via the melt blending method in the literature have a certain reinforcing effect, but the ductility improves slightly (elongation at break around 150%). So, it is of great significance to prepare composites with high ductility without losing tensile strength.

Polyethylene glycol (PEG) is non-toxic, biocompatible and easily dissolved in water and various organic solvents [21]. More importantly, it can be used as an effective plasticizing material for PLA [22,23]. However, the tensile strength of PLA could be decreased by about 50% by adding PEG only [24]. On the other hand, the surface modification of GO with PEG (GO-g-PEG) has been extensively studied in drug delivery application [25]. In order to achieve a better toughening effect, GO is required to graft more PEG. Unfortunately, the carboxyl group of GO is only distributed on the edge of the nanosheet, while the hydroxyl and epoxy groups are distributed on the surface [26,27].

Therefore, Williamson synthesis was used to convert some of the hydroxyl groups of GO into carboxyl groups, resulting in more a reactive position of GO for hydroxyl groups of PEG [28,29]. Here, the optimum process of carboxylation and the effect of carboxylated graphene oxide (GC) grafted with PEG (GC-g-PEG) content on PLA were studied. In order to make a clear comparison, the effect of GO-g-PEG on PLA was also discussed. To the best of our knowledge, the best process for the carboxylation of GO and the effect of compounding with PLA has not been investigated.

# 2. Experimental section

## 2.1. Materials

PLA (trade name 4032D) was purchased from Nature Works LLC (USA). Natural graphite powder (325 meshes), 4-dimethylaminopyridine (DMAP, ≥98%), dimethylcyclohexylcarbodiimide (DCC, ≥98%), *N,N*-dimethylformamide (DMF, 99.8%), NaOH (≥98%, pellets), ethanol (≥99.5%) and PEG with a weight-average molecular weight of $2 \times 10^2 \, \text{g mol}^{-1}$ were delivered by Aladdin Industrial Corporation, China. Chloroacetic acid (≥99.5%) was purchased from the Tianjin Damao Chemical Reagent Factory. Deionized water (DI) was used for all experimental water. All chemicals were used as received without any further purification.

## 2.2. Preparation of carboxylated graphene oxide

GO was synthesized from natural graphite powders via a modified Hummer's method [30,31]. Subsequently, 1 g of NaOH was added to 10 ml of GO aqueous suspension at a concentration of $2 \, \text{mg ml}^{-1}$, maintaining the mass ratio as 50 : 1. After that, chloroacetic acid was added to convert the hydroxyl group into a carboxyl group. The carboxylation procedure of GO is illustrated in scheme 1.

In order to investigate the effect of chloroacetic acid content on the graft ratio of GO, a series of GCs with the same NaOH/GO feed ratio were prepared with different amounts of chloroacetic acid (varying

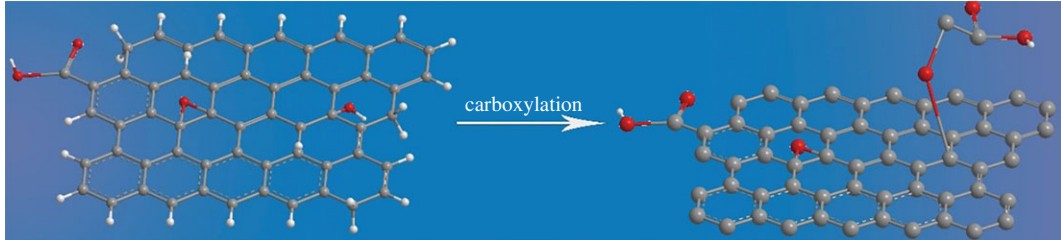

**Scheme 1.** Carboxylation of graphene oxide.

from 0, 25, 50 and 75 times to the content of NaOH). For abbreviation, the GCs were denoted as 'GC$x$', of which the $x$ represents the aim loading ratio of chloroacetic acid to NaOH.

## 2.3. Preparation of carboxylated graphene oxide grafted with polyethylene glycol, and graphene oxide grafted with polyethylene glycol

With the aid of ultrasonication, 100 mg of dried GC was added into 500 ml of DMF, forming a stable GC/DMF solution. Then, 0.5 g of PEG was added and stirred for 15 min before adding DCC and DMAP. The mixture was stirred at room temperature for 3 days, followed by an exhaustive dialysis process against DI water to remove all small molecules and unreacted PEG. The obtained composites were denoted as GC-g-PEG. In comparison, GO-g-PEG was synthesized via the same procedure.

## 2.4. Preparation of polylactic acid nanocomposites

The masterbatch was prepared by the solution method to ensure the dispersion of GC-g-PEG and GO-g-PEG. Taking PLA/GC-g-PEG (100 : 0.1, w/w) as an example, 40 mg of GC-g-PEG was added in 40 ml of ethanol, followed by a bath sonication procedure for 60 min. Meanwhile, 2 g of PLA was dissolved in 20 ml of chloroform with a stirring speed of 45 rpm until dissolved completely. Then, the ethanol/GC-g-PEG suspension was poured into the chloroform solution of PLA, and excessive ethanol was added until no precipitation could be observed. The solution was then transferred to a PTFE evaporator to volatilize the liquid phase. After completely evaporating the ethanol solvent in a vacuum oven at 60°C, the masterbatch can be obtained and cut into small pieces.

Prior to blending, PLA was dried in an oven at 80°C, while GC-g-PEG was dried in a vacuum oven at 50°C for at least 12 h. Melt blending was then operated using a HAKKE RHEOMIX (Thermo Fisher Scientific, Germany) at 175°C with a screw speed of 55 rpm for 5 min. Then, the mixture was made into a dumbbell shape via a hot-pressing procedure with the vacuum laminating machine under the condition of 5 MPa and 175°C for 5 min. For clarification, the nanocomposites are labelled as PLA/GC-g-PEG $x$, where $x$ represents the mass fraction of GC-g-PEG. Similarly, PLA/GO-g-PEG 0.4, PLA/GO and PLA/GC were prepared by the same method for comparison.

## 2.5. Characterization

The chemical structure of the GO and GCs was analysed by a Spectrum GX Fourier transform infrared (FTIR) system (PerkinElmer, USA) with a resolution of 4 cm$^{-1}$ and 32 scans in the wave number range from 400 to 4000 cm$^{-1}$ at room temperature. X-ray diffraction (XRD) measurement was carried out by a (Bruker, Germany) X-ray diffractometer at room temperature. The diffracted intensity of Cu K$\alpha$ radiation ($\lambda = 1.542$ Å) was recorded at a scan rate of 5° min$^{-1}$ from 5° to 30°. The graft ratio of modified GO was calculated from thermogravimetric analysis (TGA) using a NETZSCH-TA4/TG209 thermoanalyser (Germany), which was conducted under N$_2$ atmosphere from ambient temperature to 750°C (10°C min$^{-1}$). Raman spectra were recorded on a RM 2000 Microscopic confocal Raman spectrometer (Renishaw PLC, UK) using a 523 nm laser beam. X-ray photoelectron (XPS) spectra were obtained with a XSAM800 X-ray spectrometer (Kratos, Japan) with double anode, operating in an FAT mode, with a pass energy of 20 eV and a power of 120 W. Al K$\alpha$ X-ray (1486.6 eV) was chosen as the source of radiation.

The crystallization behaviour of PLA, PLA/GO-g-PEG and PLA/GC-g-PEG nanocomposites was studied by using differential scanning calorimetry (DSC; NETZSCH-Q2000, Germany) measurements on a DSC instrument. Under the protection of nitrogen, the samples were firstly heated up to 200°C at a rate of 10°C min$^{-1}$ and held at 200°C for 3 min to remove the thermal history. Then, the samples

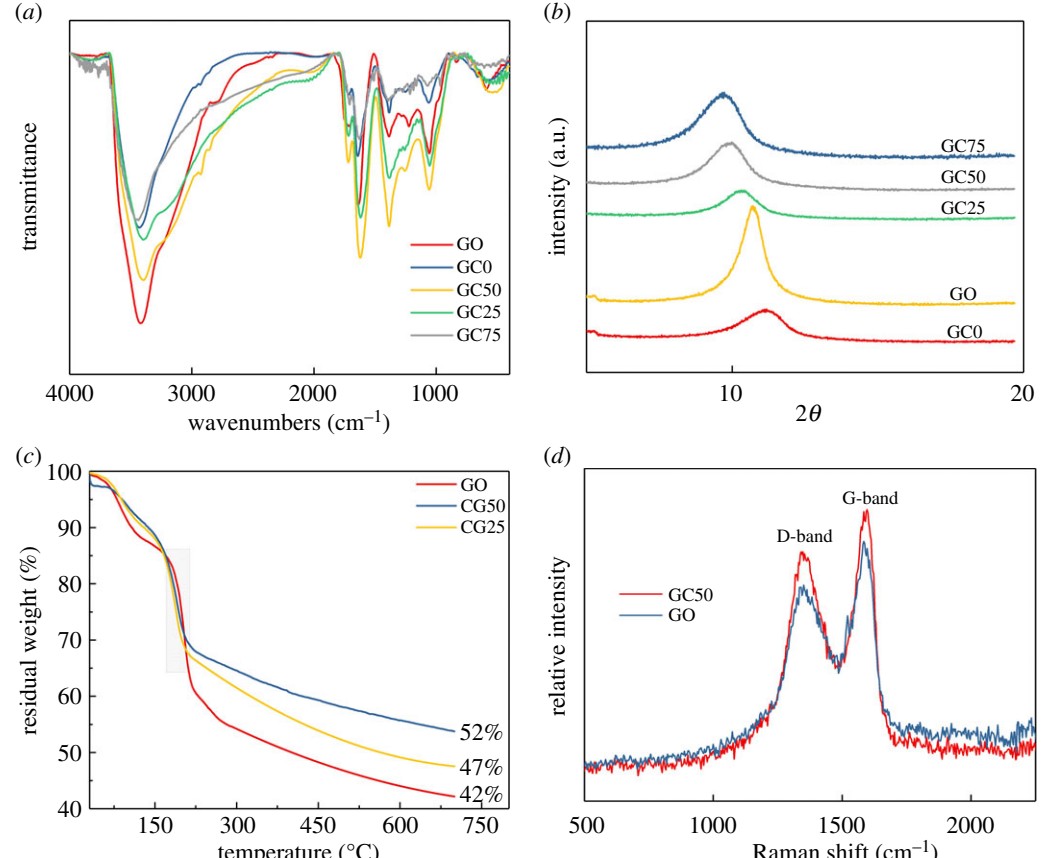

**Figure 1.** Characterizations of GO and GCs: (a,b) FTIR and XRD of GO and GCs with different contents of chloroacetic acid, respectively. (c) TGA curves of GC50 and GC25. (d) Raman spectra of GO and GC50.

were cooled down to 20°C at a cooling rate of 10°C min$^{-1}$. Finally, the samples were again heated up to 200°C at a rate of 10°C min$^{-1}$. Glass transition temperature ($T_g$), cold crystallization temperature ($T_{cc}$) and melting point ($T_m$) are determined by the secondary temperature curve. The thermal stability of nanocomposites can also be evaluated by TGA, which is conducted with the scan range from 30° to 750° at a constant heating rate of 10°C min$^{-1}$ and continuous nitrogen flow.

The tensile properties of the samples were measured using the universal electronic tensile machine (CMT-5104, China) with reference to ASTM D638. The tensile rate was 5 mm min$^{-1}$, the load cell size is 10 kN and at least five splines were tested in each group. Scanning electron microscopy (SEM) was conducted using a Hitachi S-5500 scanning electron microscope (Japan) at an acceleration voltage of 15 kV, and all the samples were sputter-coated with gold. The transmission electron microscopy (TEM) characterization was performed using a FEI Talos f200X electron (USA) microscope at an operating voltage of 220 kV.

# 3. Results and discussion

## 3.1. Characterization of carboxylated graphene oxide, graphene oxide grafted with polyethylene glycol, and carboxylated graphene oxide grafted with polyethylene glycol

In order to optimize the condition for carboxylation, different amounts of chloroacetic acid (e.g. 0, 25, 50 and 75 times to the content of alkali) were examined by applying them through the identical procedure with the same alkali concentration. The synthesized GC$x$ and GO were characterized via FTIR, XRD, TGA, XPS, Raman and TEM. Figure 1$a$ shows the FTIR spectra of GO and GC$x$ dispersed in high-quality and non-scattering potassium bromide (KBr) discs. Typical characteristic peaks of GO at 1720 cm$^{-1}$ (C=O in carboxylic groups), 1617 cm$^{-1}$ (C=C in aromatic ring), 1380 cm$^{-1}$ (C–O in carboxylic groups), 1249 cm$^{-1}$ (hydroxyl C–OH) and 1046 cm$^{-1}$ (C–O–C in epoxide) were observed, as displayed in figure 1$a$ [32]. Interestingly, reflection at 1720 cm$^{-1}$ over GC50 and GC25, corresponding to C=O in the carboxyl group, is sharper than that observed over GO. Conversely, reflection at 1720 cm$^{-1}$ over GC75 and GC0 is weaker than that observed over GO. Meanwhile, the peak at

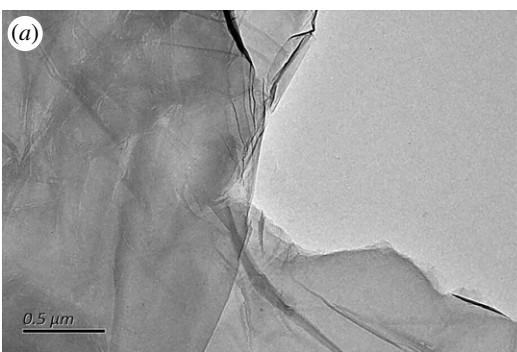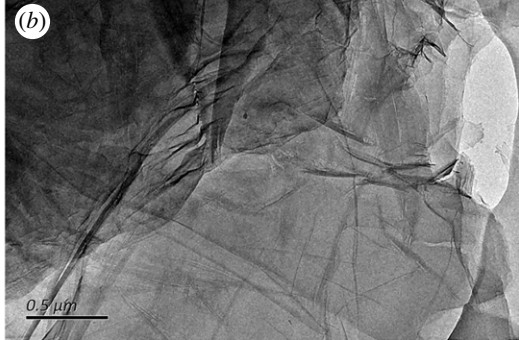

**Figure 2.** TEM images of GO (*a*) and GC50 (*b*).

3411 cm$^{-1}$ obtained over GC$x$ is weakened in comparison to that detected over the original GO, indicating that some of the hydroxyl groups are converted into carboxyl groups [33,34].

In addition, as shown in figure 1*b*, after graphite is intercalated with sulfate ions, the distance between the sheets increased to 8.17 Å. When the chloroacetic acid is grafted onto the GO surface, spacing expansion between the layers is also noted. For instance, when the content of chloroacetic acid is 25 times that of GO, the diffraction peak shifts from 10.74° to 10.44°, and the d-spacing increases from 8.17 to 8.41 Å. The d-spacing further increases to 8.9 Å for GC50. However, when chloroacetic acid content keeps increasing, i.e. GC75, the enhancement of layer spacing cannot be observed. In addition, it is interesting to note that the diffraction peak over GC0 shifts towards the right and the d-spacing decreases to 7.81 Å in comparison to that observed over GO. On the other hand, when sodium hydroxide is added without chloroacetic acid, the interlayer spacing is further reduced, which may be owing to the reduction reaction of GO [35].

It has been established that the hydroxyl groups on GO sheets begin to decompose at 110°C and all hydroxyl groups disappear at 150°C, the remaining oxygen-containing functional groups gradually decompose at 150–230°C [36]. Figure 1*c* displays the thermal decomposition behaviours of the GO and GCs. As expected, two steps of weight loss over GO can be observed, which happens at 90–150°C and 150–230°C, respectively. The first stage can be attributed to the loss of the surface hydroxyl group and adsorbed water, and the second stage is mainly owing to the thermal decomposition of the relatively stable oxygen-containing functional groups in the GO structure. In comparison, one step of weight loss over GCs is displayed. The platform at 90–150°C disappeared, demonstrating a decrease in mass loss at 90–150°C, while an increased mass loss at 150–230°C is detected. This further suggests that the hydroxyl groups of GO are partially converted to carboxyl groups. In addition, the grafting ratio of carboxyl groups is calculated to be approximately 5–10%. The corresponding Raman spectra of GO and GC50 are displayed in figure 1*d*. It can be seen that both Raman spectra contain the D- and G-bands at around 1341 cm$^{-1}$ and 1596 cm$^{-1}$, respectively. The intense D-band corresponded to the vibration of sp$^3$-bonded carbon atoms and related to the structural defects of GO, while the G-band was reflective of the in-plane vibration of sp$^2$-bonded carbon atoms. The intensity ratio of the D- and G-bands ($I_D/I_G$) of GO is 1.07 and that of GC50 is 1.16, which indicates that the ordered structure was further disrupted during the modification process. This is probably owing to the fact that the graft reaction of the chloroacetic acid segments can increase the disorder of the system.

The morphologies of GO and GC50 were investigated by TEM. GO showed a typical wrinkle layer structure and high transparency (figure 2*a*). The wrinkles of GC increased and transparency became lower (figure 2*b*). After reaction with chloroacetic acid, the morphology was apparently maintained.

In order to quantify the conversion of carboxyl functional group, XPS was performed to analyse the carboxyl content of GO and GC50, as shown in figure 3. From the XPS spectra of both GO and GC50, five peaks at 284.5 eV, 285.2 eV, 286.6 eV, 287.6 eV and 288.9 eV could be observed, which reflect C–C, C=C, C–O, C=O, O=C–O, respectively [26,37]. It is noted that the intensity of the C–O peak of GO is significantly higher than that of GC50, i.e. 45.8 versus 32.9%. Furthermore, the peak area of O=C–O over GO is 6%, while 17% is obtained for that of GC50. This result is consistent with FTIR, XRD and TGA results. This suggests that chloroacetic acid is grafted onto GO through a substitution reaction, generating approximately 11% carboxyl functional groups on the GO surface, leading to more reaction sites for PEG.

Herein, GC50 was selected to be esterified with PEG. In order to check whether grafting reactions were conducted, FTIR, TGA and XRD were conducted. FTIR spectra, TGA thermogram and XRD pattern of the GO, GC-g-PEG and GO-g-PEG are presented in figure 4.

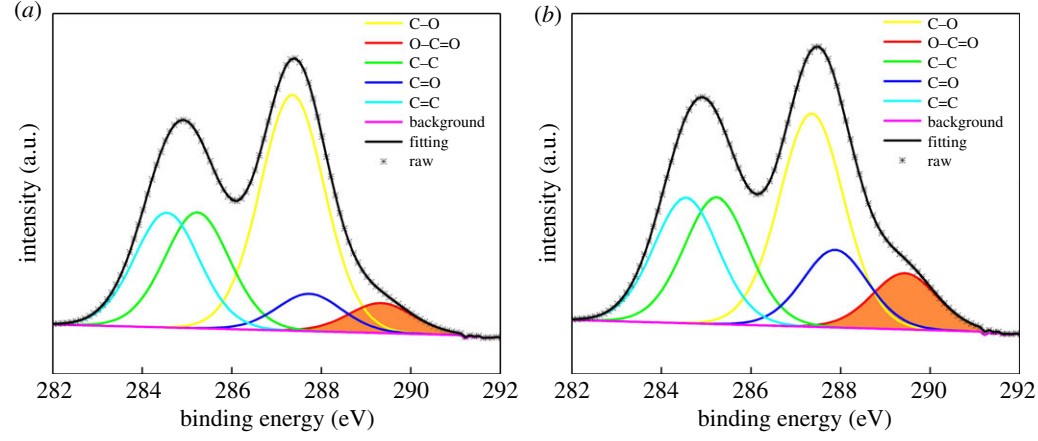

**Figure 3.** XPS C1s spectra of GO (*a*) and GC50 (*b*).

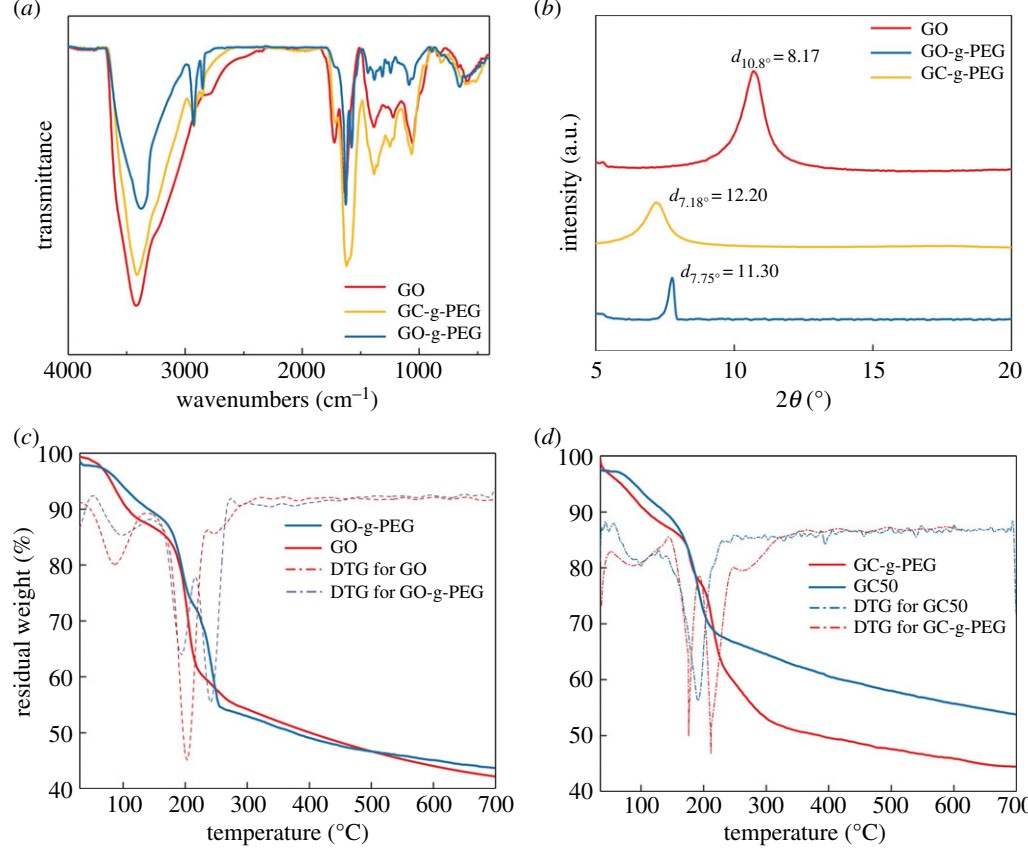

**Figure 4.** FTIR, TGA and XRD patterns of GO, GO-g-PEG and GC-g-PEG: (*a*) FT-IR, (*b*) XRD and (*c,d*) TGA.

FTIR spectra of GC-g-PEG and GO-g-PEG dispersed in high-quality and non-scattering KBr discs are shown in figure 4*a*. Two new peaks at 2928 cm$^{-1}$ and 2838 cm$^{-1}$ were observed over GC-g-PEG and GO-g-PEG, respectively, corresponding to –CH$_2$ of the PEG, indicating that PEG was successfully grafted onto GO and GC by the esterification reaction [38]. This can also be proved by the shift of the peak from 1720 to 1730 cm$^{-1}$. XRD patterns of the original GO as well as GO-g-PEG and GC-g-PEG are displayed in figure 4*b*. After modified by PEG through the esterification reaction, two reflections at 7.75° and 7.18° over GO-g-PEG and GC-g-PEG were detected, which corresponds to d-spacing of 11.30 and 12.20 Å, respectively. This further implies that PEG is grafted on GO and GC successfully, which is consistent with the FTIR results. Figure 4*c,d* demonstrates the thermal decomposition behaviours of the GO-g-PEG and the original GO as well as GC-g-PEG and GC50, respectively. In comparison to the original GO and GC50, there is an extra weight loss step observed over GO-g-PEG and GC-g-PEG at 170–250°C. This can be attributed to the thermal decomposition of the PEG

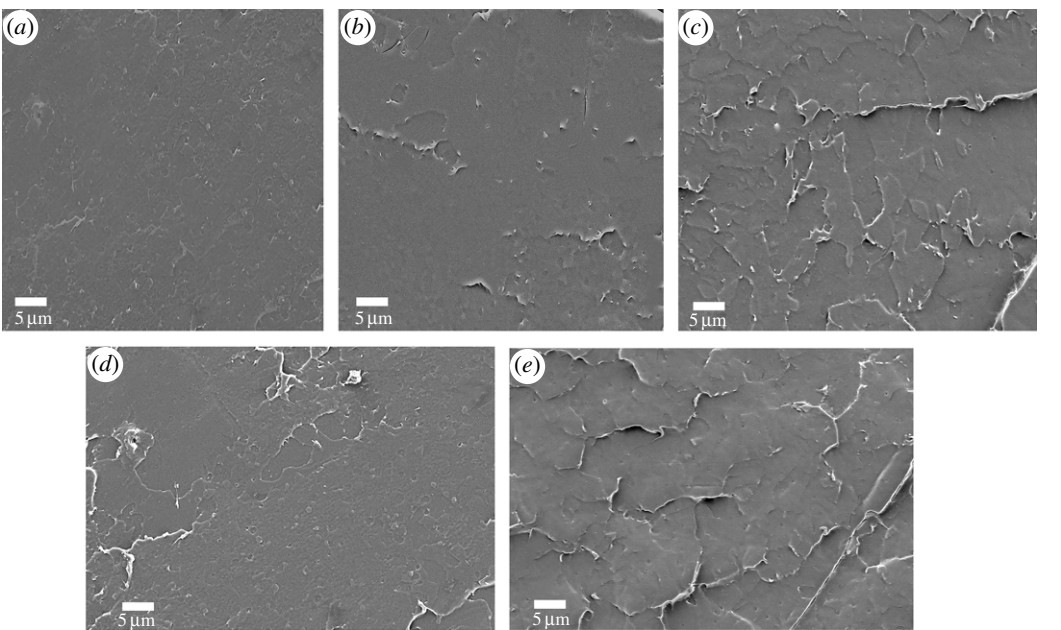

**Figure 5.** SEM images of cryofractured cross-section of (*a*) pure PLA, (*b*) PLA/GC-g-PEG 0.3, (*c*) PLA/GC-g-PEG 0.4, (*d*) PLA/GO-g-PEG 0.4 and (*e*) PLA/GO-g-PEG 0.5 nanocomposites.

molecular chain. Note, the processing temperature of PLA is around 170°C, for which GO-g-PEG and GC-g-PEG can be processed at this temperature.

## 3.2. Structure and morphology of polylactic acid nanocomposites

Good dispersion of nanofillers in the polymer matrix is of great essence for high performance of nanocomposites [15,39]. SEM was applied to clarify the dispersion of modified GO nanofillers in PLA. Cryofractured cross-section of pure PLA, PLA/GC-g-PEG 0.3, PLA/GC-g-PEG 0.4, PLA/GO-g-PEG 0.4 and PLA/GO-g-PEG 0.5 is shown in figure 5. From figure 5*a*, pure PLA exhibits a relatively smooth fracture surface with no plastic deformation. However, when 0.4% GO-g-PEG was added, as illustrated in figure 5*d*, the PLA surface became rough and dense, and no obvious interface debonding sign was observed. However, cracking of the surface was observed, showing limited interactions between PLA and GO-g-PEG, indicating only a partial compatibility between them. On the other hand, when 0.3% GC-g-PEG was applied, the boundary between the PLA matrix and the GC-g-PEG becomes indistinguishable (figure 5*b*). This might be rationalized in terms that more PEG is grafted onto the GC in comparison to that achieved over GO, leading to stronger interfacial interaction between GC-g-PEG and PLA, which is completely infiltrated by PLA. Unfortunately, when 0.5% GO-g-PEG and 0.4% GC-g-PEG are added, more cracks and agglomeration can be observed, respectively. The aggregation of nanosheets can be attributed to the inevitable van der Waals force.

The dispersion of GC-g-PEG was further evaluated by the TEM, which is shown in figure 6. The TEM image of the PLA/GC-g-PEG 0.3 indicates that most of the GC-g-PEG are exfoliated and well embedded in the PLA matrix, showing a molecular level of dispersion.

## 3.3. Crystallization behaviour of the polylactic acid nanocomposite

As a semi-crystalline polymer, the crystallization behaviour of PLA plays a guiding role in the mechanical properties to some extent [40]. The DSC trace obtained for the pure PLA and PLA/GO-g-PEG 0.4, as well as PLA with the different content of GC-g-PEG, is shown in figure 7. Table 1 gives the typical thermal properties, including $T_g$, $T_{cc}$, $T_m$, the melting enthalpy ($\Delta H_m$) and the crystallinity ($X_c$). Furthermore, the DSC heating curves of PLA nanocomposites with the different content of GO-g-PEG are shown in the electronic supplementary material, figure S2.

Increasing the loading of GC-g-PEG from 0% to 0.3%, $T_g$ and $T_{cc}$ of PLA dropped from 62.3°C and 122.3°C to 56.2°C and 101.4°C, respectively. Similarly, when the GO-g-PEG content was 0.4%, $T_g$ and $T_{cc}$ also decreased to 56.6°C and 102.5°C, respectively. This is because the grafted PEG can be used as a

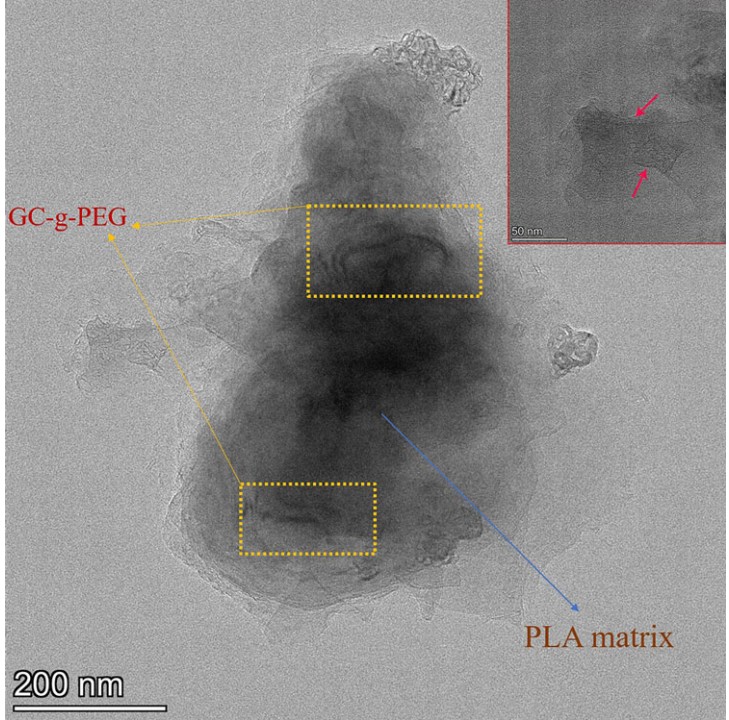

**Figure 6.** TEM images of PLA/GC-g-PEG 0.3 nanocomposite.

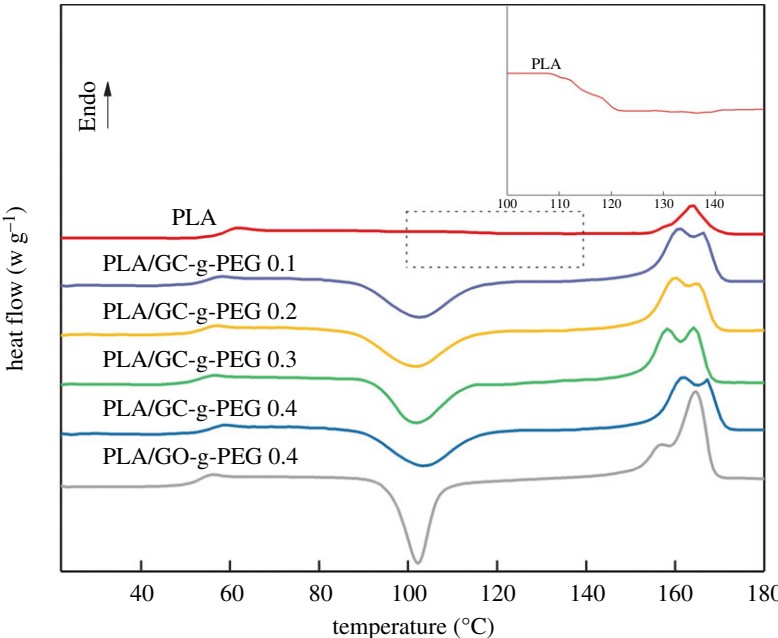

**Figure 7.** DSC heating curves of pure PLA and PLA/GO-g-PEG 0.4 as well as PLA/GC-g-PEG x.

low-molecular-weight plasticizer, which increases the lubricity of molecular chains and the free volume of molecular chains. Therefore, less energy is needed to move the molecular chain in comparison to pure PLA, which could result in a decrease in the $T_g$ of the entire nanocomposite. However, when the loading of GC-g-PEG is further raised to 0.4%, the values of $T_g$ and $T_{cc}$ stop increasing, which may be owing to the inevitable agglomeration of the nanofillers, partially hindering the mobility of the molecular chain. The process of cold crystallization and crystal melting of pure PLA is shown in figure 7. Owing to the rigidity of the pure PLA molecular chain and the interaction of intermolecular hydrogen bonds, the temperature for cold crystallization is high (122.3°C), the range is wide. The addition of GC-g-PEG and GO-g-PEG reduces the cold crystallization temperature of PLA to around 102°C and shortens the temperature

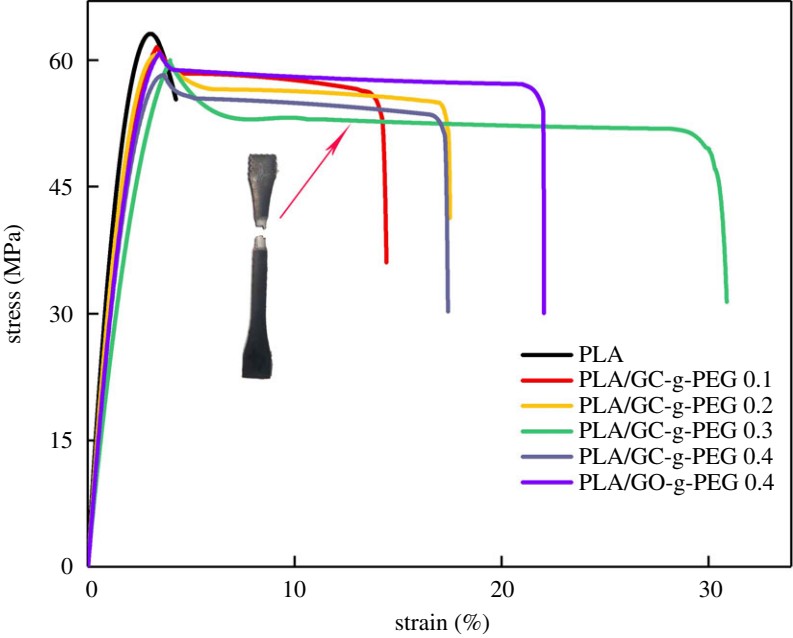

**Figure 8.** Typical nominal stress–strain curves for pure PLA and PLA/GO-g-PEG 0.4 as well as PLA/GC-g-PEG x.

**Table 1.** DSC parameters for pure PLA and PLA/GO-g-PEG 0.4 as well as PLA/GC-g-PEG x including $T_g$, $T_{cc}$, $T_m$, $\Delta H_m$ and $X_c$.

| samples | $T_g$ (°C) | $T_{cc}$ (°C) | $T_{m1}$ (°C) | $T_{m2}$ (°C) | $\Delta H_m$ (°C) | $X_c$ (%) |
|---|---|---|---|---|---|---|
| PLA | 62.3 | 122.3 | — | 163.3 | 15.1 | 16.2 |
| PLA/GC-g-PEG 0.1 | 59.3 | 103.3 | 162.3 | 167.3 | 37.9 | 40.7 |
| PLA/GC-g-PEG 0.2 | 57.4 | 102.2 | 160.4 | 165.4 | 38.0 | 40.9 |
| PLA/GC-g-PEG 0.3 | 56.2 | 101.4 | 158.2 | 164.2 | 40.9 | 44.1 |
| PLA/GC-g-PEG 0.4 | 58.4 | 103.4 | 161.4 | 166.4 | 37.6 | 40.5 |
| PLA/GO-g-PEG 0.4 | 56.6 | 102.5 | 156.7 | 164.6 | 35.2 | 37.8 |

range of cold crystallization. It can also be observed that, in comparison to the single peak observed over pure PLA, the melting process of PLA/GC-g-PEG x and PLA/GO-g-PEG 0.4 exhibits a distinct bimodal phenomenon. The reason for this phenomenon is that an unstable crystal is formed at a low temperature (about 160°C), with increasing the temperature, thermodynamically, the unstable crystal regions will be partially rearranged to stabilize the crystal structure [41].

In addition, the $X_c$ of PLA can be determined by comparing the obtained $\Delta H_m$ of PLA with its theoretical melting enthalpy (93 J g$^{-1}$) [42]. The $X_c$ of pure PLA is only 16.2%, as shown in table 1. However, when the GC-g-PEG content is 0.3%, the highest $X_c$ of 44.1% over PLA is achieved. With the content increasing, the $X_c$ decreases to 40.5%, which is ascribed to the agglomeration of GC-g-PEG. The increase of $X_c$ can be attributed to the fact that the mobility of PLA molecular chains can be improved by the plasticizing effect of PEG. Moreover, GO and its derivatives, as crystallization heterogeneous nucleating agents, could also promote the crystallization ability of PLA [38]. The same phenomenon can be seen from table 1 that when the GO-g-PEG content is 0.4%, the $X_c$ of PLA is 37.8%. More DSC curves with different GO-g-PEG contents are presented in the electronic supplementary material, figure S2 and table S1.

## 3.4. Mechanical properties of the polylactic acid nanocomposite

It has been established that the mechanical performance of PLA could be improved when graphene-based derivatives are introduced, which is attributed to the large surface area and comparatively high modulus [32,43]. Regarding this, the mechanical properties of neat PLA, PLA/GO-g-PEG 0.4 and

**Table 2.** Tensile properties of pure PLA and PLA/GO-g-PEG 0.4 as well as PLA/GC-g-PEG x.

| samples | tensile strength (MPa) | elongation at break (%) | modulus (GPa) | $W_b$ (MJ m$^{-3}$) |
|---|---|---|---|---|
| PLA | 63 | 4 | 2.1 | 20.1 |
| PLA/GC-g-PEG 0.1 | 61 | 13 | 1.8 | 76.9 |
| PLA/GC-g-PEG 0.2 | 60 | 18 | 1.7 | 93.7 |
| PLA/GC-g-PEG 0.3 | 59 | 31 | 1.5 | 155.8 |
| PLA/GC-g-PEG 0.4 | 58 | 17 | 1.6 | 89.3 |
| PLA/GO-g-PEG 0.4 | 60 | 22 | 1.8 | 120.8 |

PLA/GC-g-PEG x were estimated by the tensile test. The typical stress–strain curves are presented in figure 8. It can be noted that the neat PLA is rigid and brittle, which possesses strong tensile modulus and strength (2.1 GPa and 63 MPa, respectively). Unfortunately, low elongation at break (4%) is observed. When GO-g-PEG 0.4 is added, the elongation at break increased to 22%, and only a slight decrease in tensile modulus and strength is observed (1.8 GPa and 60 MPa, respectively). For comparison, more mechanical properties and graphs with different GO-g-PEG contents are given in the electronic supplementary material, table S2 and figure S3. On the other hand, it is known that there are more carboxyl groups remaining over GC, which is mainly owing to the carboxylation modification of GO. This leads to the fact that more PEG molecule chains can be grafted on the surface of GC. Therefore, higher elongation at break could be obtained, of which 31% was achieved over PLA/GC-g-PEG 0.3. However, an unfortunate decrease in tensile modulus and strength was observed (1.5 GPa and 59 MPa, respectively), which might be rationalized in terms that more flexible PEG chains were grafted. By the way, the practical application of PLA will not be affected by this reduction. Owing to the incompatibility between the PLA and GO/GC, the mechanical performance of them is unsatisfactory (electronic supplementary material, table S3 and figure S4).

Generally speaking, tensile toughness is a measurement of the energy absorbed by a material during its tensile fracture. The absorbed energy ($W_b$) can be calculated from the integrated area under the stress–strain curve, which is $W_b = \int_0^\varepsilon \sigma \mathrm{d}\varepsilon$ [44]. Tensile properties of pure PLA and its nanocomposites are listed in table 2. With increasing the content of added GC-g-PEG from 0 to 0.3%, the $W_b$ rises from 20.1 to 155.8 MJ m$^{-3}$, which is 11 times higher than that of pure PLA. When the content of GO-g-PEG is 0.4%, the $W_b$ rises to 120.8 MJ m$^{-3}$. However, when the content of added GC-g-PEG is further increased to 0.4%, the $W_b$ decreases to 89.3 MJ m$^{-3}$, which can be attributed to the inevitable agglomeration of nanofillers.

Interestingly, when we analysed the data collected from table 2 and electronic supplementary material, table S2, we find that when the content of the nanofiller is below 0.4 wt%, the PLA nanocomposites fabricated by the GC-g-PEG demonstrated a good ductility, being superior to those made by the GO-g-PEG with the same content. As the carboxyl groups present in the GC can afford more active sites, so more PEG will be grafted onto the GC, resulting in better performance under the same content. For instance, when the content of GC-g-PEG was 0.2 wt%, the elongation at break of PLA was 18%, which was equal to the value of 0.3 wt% GO-g-PEG, while the same content of GO-g-PEG can only achieve the value of 16%. Moreover, when the content of GO-g-PEG was 0.4 wt%, the elongation at break of PLA has just increased to 22%, but it can reach 31% when the content of GC-g-PEG was 0.3 wt%. Unfortunately, the elongation at break of PLA tended to decrease following the increasing amount of GC-g-PEG. Anyway, when GO was carboxylated, better results can be achieved by simply loading less nanofillers on PLA.

The mechanical properties of polymeric nanomaterials are largely influenced by the dispersibility of the nanofillers and the compatibility with the matrix resin [15]. It is known that both GO-g-PEG and GC-g-PEG can be dispersed in the PLA uniformly (figure 5). This could be explained as follows: when PEG is grafted onto GO and its derivatives, as a small molecule, it can penetrate into the interior of the PLA molecule to increase the intermolecular lubricity and reduce the internal deformation resistance. Thus, the strong van der Waals force and hydrogen bond between the PLA molecules can be overcome, increasing the compatibility of the material and nanofillers as well as the elongation at break. Consequently, the ductility of PLA nanocomposites is improved while maintaining their own strength because of the promising dispersibility and enhanced interfacial interactions.

On the other hand, we also compared the elongation at break as well as tensile strength of GC-g-PEG on the PLA matrix in this work with other graphene-type nanofillers (figure 9), from which one can

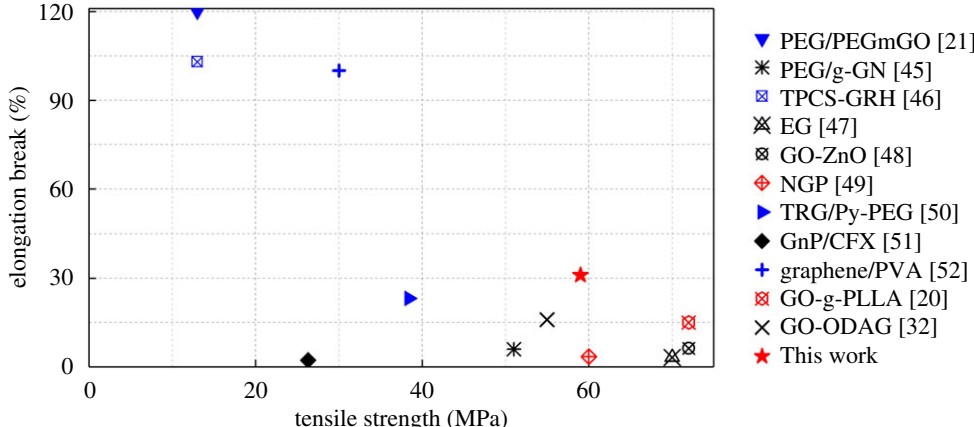

**Figure 9.** Comparison of the toughening effect of graphene-type nanofillers on PLA [20,21,32,45–52].

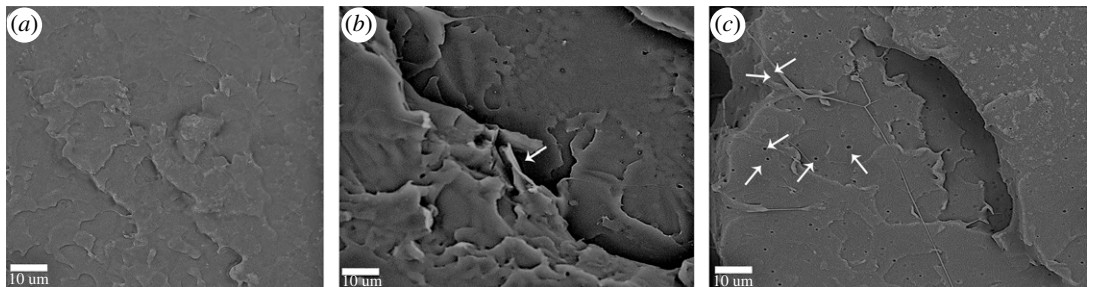

**Figure 10.** The SEM micrographs of the nanocomposites tensile-fractured surfaces: (*a*) pure PLA, (*b*) PLA/GO-g-PEG 0.4 and (*c*) PLA/GC-g-PEG 0.3.

observe that the toughening as well as the reinforcing effect of GC-g-PEG on PLA in this study was superior to most of the graphene-type nanofillers/PLA systems.

Figure 10 shows the SEM graphs of the tensile fracture surfaces of pure PLA, PLA/GO-g-PEG 0.4 and PLA/GC-g-PEG 0.3. It is well known that cracks are the main fracture mechanisms in rigid polymers [53]. If there is no effective termination mechanism, it will finally develop into catastrophic cracks. However, stressed concentration at the interface between PLA and GO is caused owing to their inherent incompatibility. Eventually, the interface fails, leading the cracks to ultimately propagate, so the brittle fracture finally occurs [54]. Therefore, it can be seen from figure 10*a* that pure PLA exhibits a relatively smooth fracture surface, revealing a typical behaviour of brittle fracture. When 0.4% GO-g-PEG was applied, the fracture becomes rough and pull-out can be observed, leading to the promotion of PLA shear yielding (figure 10*b*). When 0.3% of GC-g-PEG was added, the surface became rougher, and more pull-out and even microvoids were observed, indicating considerable tensile energy dissipation and a stronger shear yielding phenomenon (figure 10*c*).

From the above discussion, it is realized that the addition of GC-g-PEG or GO-g-PEG can improve the interface compatibility between GC/GO and PLA, and then the mechanical properties of PLA will be improved. This can be attributed to the fact that PEG acts as a transmitter during the stress transmission at the interface between PLA and GO. When external stress is applied to the material, the PEG at the interface can induce plastic deformation, resulting in the slippage of the PLA molecular chain. The obvious microvoids are demonstrated in figure 10*c*, which can be rationalized in terms that the nanofillers will rupture under tensile loading. This leads to a decline in the capacity of load-carrying for nanofillers, in which the original load will be transferred to the surrounding matrix or unfragmented nanoparticles, thus increasing the local deformation of the surrounding matrix. It has been reported that energy dissipation and shear yielding of matrix occur once voids are created [55,56]. Therefore, the tensile toughening mechanism of PLA can be attributed to the fact that GO-g-PEG or GC-g-PEG can form voids with a hollow cylindrical and microscale shape during the tensile process, which promotes shear yield rather than cracks [57]. The SEM micrographs (low magnification) of the pure PLA and PLA/GC-g-PEG 0.3 nanocomposites tensile fracture surfaces can also be seen in the electronic supplementary material, figure S5.

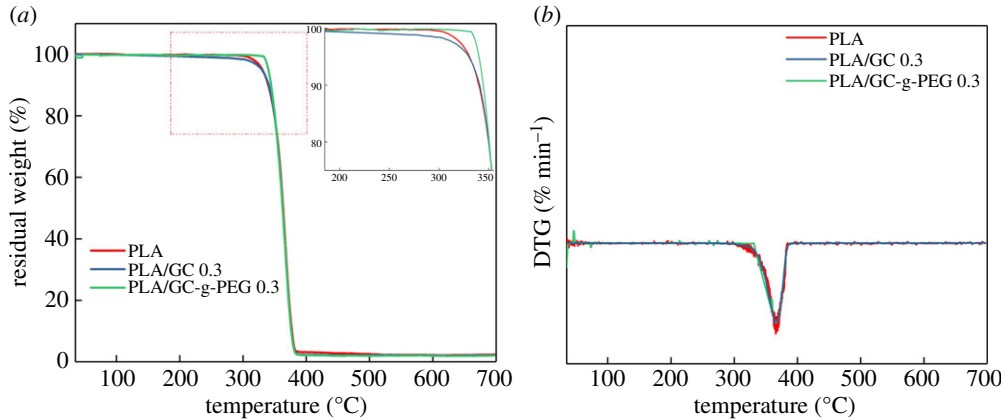

**Figure 11.** (*a*) TGA and (*b*) DTA curves for neat PLA, PLA/GC 0.3 and PLA/GC-g-PEG 0.3 nanocomposite.

**Table 3.** TGA parameters for pure PLA, PLA/GC 0.3 and PLA/GC-g-PEG 0.3 nanocomposite determined from the TGA curves.

| samples | $T_5$ (°C) | $T_{50}$ (°C) | $T_{max}$ (°C) |
|---|---|---|---|
| PLA | 330 | 363 | 365 |
| PLA/GC 0.3 | 329 | 362 | 365 |
| PLA/GC-g-PEG 0.3 | 343 | 364 | 367 |

## 3.5. Thermal stability of the polylactic acid nanocomposite

Some studies have reported that thermal properties of the polymer matrix can be improved by the addition of lower loadings (≤0.5 wt%) of GO or its derivatives [24,32]. Similar results were observed in this work as well. TGA and DTA results of pure PLA, PLA/GC 0.3 and PLA/GC-g-PEG 0.3 nanocomposites are shown in figure 11. From the TGA curve, 5% ($T_5$), 50% ($T_{50}$) and maximum ($T_{max}$) decomposition temperatures can be speculated, which are shown in table 3. As observed from figure 11*a,b*, neat PLA, PLA/GC 0.3 and PLA/GC-g-PEG 0.3 nanocomposites decompose in a one-step process. Furthermore, as shown in table 3, the $T_5$ value increases from 330 to 343°C when 0.3% GC-g-PEG content is applied in pure PLA, but no obvious change in $T_{50}$ and $T_{max}$ can be observed. Interestingly, the addition of GC produced no effect on thermal properties of PLA. These indicate that the initial decomposition temperature of PLA can be increased about 13°C by adding a small amount of GC-g-PEG. The increase in thermal stability may be owing to the shielding effect of the nanosheets and the diffusion of volatile decomposition products [58]. However, as the temperature continues rising up to 350°C, the PLA is still severely degraded. The oxygen-containing functional groups on the GC-g-PEG are detached, and the reduction phenomenon can be expected, resulting in stronger thermal conductivity to PLA [59].

## 4. Conclusion

In summary, the optimum process of carboxylation and the effect of GC-g-PEG content on PLA as well as the comparison of GO-g-PEG were explored. FTIR, XRD and TGA experiments showed that the maximum grafting rate of the carboxyl group could be achieved when the concentration of chloroacetic acid was 50 times of NaOH. XPS results showed that the carboxyl group content of GC50 on the surface increased to 15% compared with that obtained over original GO (6%). Then, novel PLA nanocomposites were prepared by using GC-g-PEG as a nanofiller through the masterbatch method. Because of the interaction between PEG and PLA, the compatibility between PLA and GC was improved and uniform dispersion was achieved. The study of crystallization behaviour shows that the addition of nanofillers can accelerate the cold crystallization process and improve the crystallinity of PLA from 16.2 to 44.1%, but the Tg of PLA was decreased from 62.3 to 56.2°C. The TGA test showed that the initial decomposition temperature of PLA/GC-g-PEG 0.3 was 343°C, which is obviously higher than that of PLA (330°C). More importantly, when the addition of GC-g-PEG was 0.3%, the

elongation at break of PLA increased to 30.55%, which was 7.15 times higher than that of pure PLA, accompanied by a slight decrease in tensile strength, i.e. 7%. The good compatibility between GC-g-PEG and PLA and the uniform dispersion in the PLA matrix are mainly attributed to the improvement of the ductility of PLA while maintaining the tensile strength. When the external force loads on PLA, the PEG at the interface can bear the medium of stress transmission, and then stress yield occurs. These results show that GO or its derivatives grafted with PEG can maintain PLA strength while enhancing its own ductility as well as crystallinity and thermal stability. This work provides a promising application on medical materials, three-dimensional printing technology and so on.

Data accessibility. All the data in this investigation have been uploaded as part of the electronic supplementary material.

Authors' contributions. M.N. and J.C. designed the study. H.W. and J.L. accomplished the whole experiment. H.C., L.L. and H.Y. collected and analysed the data. X.L. and Z.C. were responsible for the synthesis of nanocomposites. H.W. interpreted the results and wrote the manuscript, H.L. helped draft the manuscript. All authors gave final approval for publication.

Competing interests. We declare we have no competing interests.

Funding. This work was supported by the National Key Research and Development Program of China (grant no. 2018YFD0400702), the National Natural Science Foundation of China (grant no. 51573169), the China Postdoctoral Science Foundation (grant no. 2016M600583), the Natural Science Foundation of Zhejiang Province (grant no. LY15E030007) and the Key Projects of Higher Education in Henan Province (grant no. 18A140008).

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
