## [Reviewer comments · Royal Society Open Science]

Review History

RSOS-192154.R0 (Original submission)

Review form: Reviewer 1

Is the manuscript scientifically sound in its present form?

No

Are the interpretations and conclusions justified by the results?

No

Is the language acceptable?

No

Do you have any ethical concerns with this paper?

No

Have you any concerns about statistical analyses in this paper?

No

Recommendation?

Major revision is needed (please make suggestions in comments)

Comments to the Author(s)

The authors synthesised polymer/functionalised GO composites using PLA as a polymer and PEG grafted on carboxylated GO (GC) or plain GO as a filler. The fabricated composites with different PEG-g-GC concentrations was characterised using SEM, TGA, XPS and FTIR. The films fabricated from these composites were characterised for their morphology, mechanical property and thermal stability. All the concepts described in this manuscript are already explored and widely published the only novelty perhaps could be the use of PLA as polymer matrix. There are no convincing evidences of improvement of nanocomposite film properties which are the premise of the manuscript here based on the data presented. Some of the suggestions are outlined below to improve the manuscript for acceptance into the journal:

1. "Unfortunately, for structural materials, there usually are some contradictions and mutual exclusion among the mechanical properties of materials, such as high intensity and low-density mutual exclusion, high-intensity and large-deformation mutual exclusion, high-intensity and high-toughness mutual exclusion" - what does this mean? This does not make any sense. Very poor language. Need a rewrite
2. "To address this issue, plenty of toughening strategies have been developed, including plasticization, copolymerization, blending with flexible polymers and adding nanofillers" - include relevant literature for individual topic
3. "Therefore, to achieve this purpose, plenty of efforts have been made on improving dispersibility and compatibility of graphene oxide (GO) in polymers" and "Good dispersion of nanofillers in polymer matrix is of great essence for high performance of nanocomposites" - should include seminal works recently published from Prof Per Zetterlund and best reviews from Prof Elodie Bourgeat-Lami
4. "So, it is of great significance to prepare composites with high ductility without losing tensile strength" - include a table comparing the ductility data obtained here and similar examples in the literature include the one's cited by the authors.
5. "followed by a sonication procedure for 60 min." - was this bath sonication or probe sonication? Specify in the manuscript
6. "chloroacetic acid content keeps increasing, i.e., GC75, the enhancement of layer spacing cannot be observed" - could this be due to the saturation of the hydroxyl groups on the GO by chloroacetic acid?
7. Include the TEM and Raman data to corroborate the claims made from XPS and XRD
8. Fig 4- the images show different magnifications. Authors should check their magnifications and scale bars. a and b look very zoomed out (low magnification) as compared to c-e.
9. Table 2 - It seems that GO-g-PEG works better than GC counterparts. First, include the reasoning in the manuscript about it and secondly, what is the point of doing carboxylation when there is a detrimental effect of it. Need to include proper explanation and discussion in the manuscript
10. It would be interesting to see the mechanical properties of PLA with just GO and GC without PEG.
11. "7a and 7b that pure PLA exhibits a smooth and nondescript fracture surface, revealing a typical behavior of brittle fracture" - not very convincing from the SEM images presented. The features in 7a looks quite similar to fig 4a. It is difficult to accept it as a fracture, instead it seems like a film drying effect. The features identified as cracks are everywhere in the film it seems and in different directions which again indicate drying effect. Judging by the two images, 4a and 7a, the scale bar in 4a seems incorrect.
12. Fig 7, how much stress was applied? Include more low magnification images to show such features are limited to the fracture site. Also include images under different stress conditions to show the increase in the 'said' voids/crack sites. These images are not convincing. It appears the features are intrinsically present in the films.
13. Fig 8, there is no evidence shown to iterate such scheme. Where are PEG-GC/GO are worm like structures or any morphology for that matter. Why would the crack initiate at the PEG/GC/GO? Wouldn't it initiate in PLA layer which has higher Tg and then preferentially

propagate through polymer layer itself. There is only 0.3% of filler here. This schematic and explanation do not support the experimental data.

14. Fig 9, this marginal reduction in T5 cannot be regarded as heat dissipation mechanism. Any filled would do the job. Recommended to include the data for just GO and GC without PEG, to show that PEG is vital for any of these phenomena described in this study. SEM images are not convincing enough to highlight the homogeneous distribution of functionalised GC/GO sheets within PLA. Can authors include any convincing evidence such as cross-section TEM?

Review form: Reviewer 2

Is the manuscript scientifically sound in its present form?

Yes

Are the interpretations and conclusions justified by the results?

Yes

Is the language acceptable?

No

Do you have any ethical concerns with this paper?

No

Have you any concerns about statistical analyses in this paper?

No

Recommendation?

Accept with minor revision (please list in comments)

Comments to the Author(s)

The authors described a process of carboxylation and content on PLA as well as the comparison of GO-g-PEG; the new materials were characterized by a variety of means. New PLA nanocomposites were prepared by using GC-g-PEG as a nanofiller. The interaction between PEG and PLA led to better dispersion. Crystallization behavior shows that the addition of nanofillers can accelerate the cold crystallization process and improve the crystallinity of PLA. The addition of GC-g-PEG at 0.3%, the elongation at break of PLA increased to 30.55%. The good compatibility between GC-g-PEG and PLA and the uniform dispersion in PLA matrix are mainly attributed to the improvement of the ductility of PLA while maintaining the tensile strength.

Overall I found the work suitably significant and comprehensive for publication.

I do have a few queries the authors should attend to:

In a number of places the English should be corrected for improved reading.

The authors claim PLA is a suitable alternate as a biobased polymer, what about cellulose, which is the most abundant on earth?

What is meant by high-intensity in the context of mechanical testing?

Mpa should be MPa throughout

hakke rheomix should be capitalized and the country of all equipment should be stated.

Review form: Reviewer 3

Is the manuscript scientifically sound in its present form?

Yes

Are the interpretations and conclusions justified by the results?

Yes

Is the language acceptable?

Yes

Do you have any ethical concerns with this paper?

No

Have you any concerns about statistical analyses in this paper?

No

Recommendation?

Accept as is

Comments to the Author(s)

The manuscript reports on the improvement of the mechanical properties of PLA by the chemical functionalization of GO with carboxylic and PEG groups. The manuscript is well written, the experiments are well performed and the conclusions are supported by experimental results. I suggest the publication of the present work in RSOS as its present form.

Decision letter (RSOS-192154.R0)

17-Jan-2020

Dear Dr Jinzhou:

Title: Polyethylene glycol grafted with carboxylated graphene oxide as novel interface modifier for polylactic acid/graphene nanocomposites

Manuscript ID: RSOS-192154

The editor assigned to your manuscript has now received comments from reviewers. We would like you to revise your paper in accordance with the referee and Subject Editor suggestions which can be found below (not including confidential reports to the Editor). Please note this decision does not guarantee eventual acceptance.

Please submit your revised paper before 09-Feb-2020. Please note that the revision deadline will expire at 00.00am on this date. If we do not hear from you within this time then it will be assumed that the paper has been withdrawn. In exceptional circumstances, extensions may be possible if agreed with the Editorial Office in advance. We do not allow multiple rounds of revision so we urge you to make every effort to fully address all of the comments at this stage. If deemed necessary by the Editors, your manuscript will be sent back to one or more of the original reviewers for assessment. If the original reviewers are not available we may invite new reviewers.

RSC Associate Editor:
Comments to the Author:
(There are no comments.)

RSC Subject Editor:
Comments to the Author:
(There are no comments.)

Reviewers' Comments to Author:
Reviewer: 1

Comments to the Author(s)
The authors synthesised polymer/functionalised GO composites using PLA as a polymer and PEG grafted on carboxylated GO (GC) or plain GO as a filler. The fabricated composites with different PEG-g-GC concentrations was characterised using SEM, TGA, XPS and FTIR. The films fabricated from these composites were characterised for their morphology, mechanical property and thermal stability. All the concepts described in this manuscript are already explored and widely published the only novelty perhaps could be the use of PLA as polymer matrix. There are no convincing evidences of improvement of nanocomposite film properties which are the premise of the manuscript here based on the data presented. Some of the suggestions are outlined below to improve the manuscript for acceptance into the journal:

1. "Unfortunately, for structural materials, there usually are some contradictions and mutual exclusion among the mechanical properties of materials, such as high intensity and low-density mutual exclusion, high-intensity and large-deformation mutual exclusion, high-intensity and high-toughness mutual exclusion" - what does this mean? This does not make any sense. Very poor language. Need a rewrite
2. "To address this issue, plenty of toughening strategies have been developed, including plasticization, copolymerization, blending with flexible polymers and adding nanofillers" - include relevant literature for individual topic
3. "Therefore, to achieve this purpose, plenty of efforts have been made on improving dispersibility and compatibility of graphene oxide (GO) in polymers" and "Good dispersion of nanofillers in polymer matrix is of great essence for high performance of nanocomposites" - should include seminal works recently published from Prof Per Zetterlund and best reviews from Prof Elodie Bourgeat-Lami
4. "So, it is of great significance to prepare composites with high ductility without losing tensile strength" - include a table comparing the ductility data obtained here and similar examples in the literature include the one's cited by the authors.
5. "followed by a sonication procedure for 60 min." - was this bath sonication or probe sonication? Specify in the manuscript
6. "chloroacetic acid content keeps increasing, i.e., GC75, the enhancement of layer spacing cannot be observed" - could this be due to the saturation of the hydroxyl groups on the GO by chloroacetic acid?
7. Include the TEM and Raman data to corroborate the claims made from XPS and XRD
8. Fig 4- the images show different magnifications. Authors should check their magnifications and scale bars. a and b look very zoomed out (low magnification) as compared to c-e.
9. Table 2 - It seems that GO-g-PEG works better than GC counterparts. First, include the reasoning in the manuscript about it and secondly, what is the point of doing carboxylation when there is a detrimental effect of it. Need to include proper explanation and discussion in the manuscript
10. It would be interesting to see the mechanical properties of PLA with just GO and GC without PEG.
11. "7a and 7b that pure PLA exhibits a smooth and nondescript fracture surface, revealing a typical behavior of brittle fracture" - not very convincing from the SEM images presented. The features in 7a looks quite similar to fig 4a. It is difficult to accept it as a fracture, instead it seems like a film drying effect. The features identified as cracks are everywhere in the film it seems and in different directions which again indicate drying effect. Judging by the two images, 4a and 7a, the scale bar in 4a seems incorrect.
12. Fig 7, how much stress was applied? Include more low magnification images to show such features are limited to the fracture site. Also include images under different stress conditions to show the increase in the 'said' voids/crack sites. These images are not convincing. It appears the features are intrinsically present in the films.
13. Fig 8, there is no evidence shown to iterate such scheme. Where are PEG-GC/GO are worm like structures or any morphology for that matter. Why would the crack initiate at the PEG/GC/GO? Wouldn't it initiate in PLA layer which has higher Tg and then preferentially propagate through polymer layer itself. There is only 0.3% of filler here. This schematic and explanation do not support the experimental data.
14. Fig 9, this marginal reduction in T5 cannot be regarded as heat dissipation mechanism. Any filled would do the job. Recommended to include the data for just GO and GC without PEG, to show that PEG is vital for any of these phenomena described in this study. SEM images are not convincing enough to highlight the homogeneous distribution of functionalised GC/GO sheets within PLA. Can authors include any convincing evidence such as cross-section TEM?

Reviewer: 2

Comments to the Author(s)

The authors described a process of carboxylation and content on PLA as well as the comparison of GO-g-PEG; the new materials were characterized by a variety of means. New PLA

nanocomposites were prepared by using GC-g-PEG as a nanofiller. The interaction between PEG and PLA led to better dispersion. Crystallization behavior shows that the addition of nanofillers can accelerate the cold crystallization process and improve the crystallinity of PLA. The addition of GC-g-PEG at 0.3%, the elongation at break of PLA increased to 30.55%. The good compatibility between GC-g-PEG and PLA and the uniform dispersion in PLA matrix are mainly attributed to the improvement of the ductility of PLA while maintaining the tensile strength.

Overall I found the work suitably significant and comprehensive for publication.

I do have a few queries the authors should attend to:

In a number of places the English should be corrected for improved reading.

The authors claim PLA is a suitable alternate as a biobased polymer, what about cellulose, which is the most abundant on earth?

What is meant by high-intensity in the context of mechanical testing?

Mpa should be MPa throughout

hakke rheomix should be capitalized and the country of all equipment should be stated.

Reviewer: 3

Comments to the Author(s)

The manuscript reports on the improvement of the mechanical properties of PLA by the chemical functionalization of GO with carboxylic and PEG groups. The manuscript is well written, the experiments are well performed and the conclusions are supported by experimental results. I suggest the publication of the present work in RSOS as its present form.

Author's Response to Decision Letter for (RSOS-192154.R0)

See Appendix A.

RSOS-192154.R1 (Revision)

Review form: Reviewer 1

Is the manuscript scientifically sound in its present form?

Yes

Are the interpretations and conclusions justified by the results?

Yes

Is the language acceptable?

No

Do you have any ethical concerns with this paper?

No

Have you any concerns about statistical analyses in this paper?

No

Recommendation?

Accept as is

Comments to the Author(s)

The authors have satisfactorily addressed the comments of this reviewer. The manuscript can be accepted for publication.

Review form: Reviewer 2

Is the manuscript scientifically sound in its present form?

Yes

Are the interpretations and conclusions justified by the results?

Yes

Is the language acceptable?

No

Do you have any ethical concerns with this paper?

No

Have you any concerns about statistical analyses in this paper?

No

Recommendation?

Accept with minor revision (please list in comments)

Comments to the Author(s)

I find the revised version of the paper a huge improvement. There are still aspects of the manuscript that can be better expressed in English, and reading by a native speaker is encouraged. But I found the authors suitably addressed my science queries, and would appear they attempted hard to also rebut comments from other reviewers which lead to an overall improvement of the paper. I can therefore recommend publication after minor corrections.

Decision letter (RSOS-192154.R1)

Dear Dr Jinzhou:

Title: Polyethylene glycol grafted with carboxylated graphene oxide as novel interface modifier for polylactic acid/graphene nanocomposites
Manuscript ID: RSOS-192154.R1

It is a pleasure to accept your manuscript in its current form for publication in Royal Society Open Science. The chemistry content of Royal Society Open Science is published in collaboration with the Royal Society of Chemistry.

RSC Associate Editor:
Comments to the Author:
(There are no comments.)

RSC Subject Editor:
Comments to the Author:
(There are no comments.)

Reviewer(s)' Comments to Author:
Reviewer: 1

Comments to the Author(s)
The authors have satisfactorily addressed the comments of this reviewer. The manuscript can be accepted for publication.

Reviewer: 2

Comments to the Author(s)
I find the revised version of the paper a huge improvement. There are still aspects of the manuscript that can be better expressed in English, and reading by a native speaker is encouraged. But I found the authors suitably addressed my science queries, and would appear they attempted hard to also rebut comments from other reviewers which lead to an overall improvement of the paper. I can therefore recommend publication after minor corrections.

Appendix A

Dear Editor and Reviewers:

At the request from your journal, we have carefully made a major revisions on our manuscript entitled "*Polyethylene glycol grafted with carboxylated graphene oxide as novel interface modifier for polylactic acid/graphene nanocomposites*" (ID: RSOS-192154) that had been submitted to *Royal Society Open Science*. The changes were highlighted with red color in the corresponding parts of the revised version.

Point-to-point responses to all the comments raised by the reviewers are summarized in the following pages. We sincerely appreciate the reviewers' instructive suggestions and your kind understanding in improving the quality of this paper. We hope that we have adequately addressed those comments.

By the way, due to the COVID-19, our laboratory and test instruments have stopped working, it is so difficult for us to supplement some experiments. So some supplementary content that we have made may not meet the requirements very well. But this paper is very important to us, we have tried our best to do it, and we do hope meet with approval. If there is any need for correction, please contact us in time.

Sincerely,

Prof. Jinzhou Chen, PhD

School of Materials Science and Engineering, Zhengzhou University,

Zhengzhou 450001, China,

E-mail: cjz@zzu.edu.cn

Responses to Review Comments

<Reviewer #1>

Comments: *The authors synthesised polymer/functionalised GO composites using PLA as a polymer and PEG grafted on carboxylated GO (GC) or plain GO as a filler. The fabricated composites with different PEG-g-GC concentrations was characterised using SEM, TGA, XPS and FTIR. The films fabricated from these composites were characterised for their morphology, mechanical property and thermal stability. All the concepts described in this manuscript are already explored and widely published the only novelty perhaps could be the use of PLA as polymer matrix. There are no convincing evidences of improvement of nanocomposite film properties which are the premise of the manuscript here based on the data presented. Some of the suggestions are outlined below to improve the manuscript for acceptance into the journal:*

Question 1. *"Unfortunately, for structural materials, there usually are some contradictions and mutual exclusion among the mechanical properties of materials, such as high intensity and low-density mutual exclusion, high-intensity and large-deformation mutual exclusion, high-intensity and high-toughness mutual exclusion" - what does this mean? This does not make any sense. Very poor language. Need a rewrite,*

RESPONSE: *In the revised script (Line 21-23 & Page 2) , we have re-written this part according to the Reviewer's suggestion.*

Unfortunately, as to PLA, because of its intrinsic brittleness and low tensile elongation at break, the application of PLA is still limited in spite of its high elastic modulus and tensile strength.

Question 2. "To address this issue, plenty of toughening strategies have been developed, including plasticization, copolymerization, blending with flexible polymers and adding nanofillers" - include relevant literature for individual topic

RESPONSE: In the revised script (Line 24-25 & Page 2) , the relevant literature has now been inserted.

Question 3. "Therefore, to achieve this purpose, plenty of efforts have been made on improving dispersibility and compatibility of graphene oxide (GO) in polymers" and "Good dispersion of nanofillers in polymer matrix is of great essence for high performance of nanocomposites" - should include seminal works recently published from Prof Per Zetterlund and best reviews from Prof Elodie Bourgeat-Lami

RESPONSE: In the revised script (Line 7 & Page 3 and Line 11 & Page 11) , the relevant literature has now been inserted.

Question 4: "So, it is of great significance to prepare composites with high ductility without losing tensile strength" - include a table comparing the ductility data obtained here and similar examples in the literature include the one's cited by the authors.

RESPONSE: In the revised script (Line15-19 &Page 18 and Fig.9 & Page 19) , we have compared the effects of different graphene-based nanofillers on the tensile properties and ductility of PLA. And have made them into the form of a graph, which is shown in Fig. 9 of the manuscript.

Question 5: "followed by a sonication procedure for 60 min." - was this bath sonication or probe sonication? Specify in the manuscript

RESPONSE: In the revised script (Line 8 & Page 5) , we are very sorry for our unclear writing. In fact, water bath sonication is the method that

we used.

Question 6: *"chloroacetic acid content keeps increasing, i.e., GC75, the enhancement of layer spacing cannot be observed" - could this be due to the saturation of the hydroxyl groups on the GO by chloroacetic acid?*

RESPONSE: *Considering the Reviewer's suggestion, we think this possibility also exists.*

But from our FTIR characterization (Fig. 1a), we can clearly see that GC50 has new characteristic peaks at 2923cm^{-1} and 2855cm^{-1} , corresponding to the characteristic peaks of methylene in chloroacetic acid. While the peak intensities of GC75 and GC25 are weak, so combining the XRD layer spacing, too much or too little chloroacetic acid is not conducive to the carboxylation process.

Anyway, after the COVID-19, we will do more experiments to perfect our assumption.

Question 7: *Include the TEM and Raman data to corroborate the claims made from XPS and XRD*

RESPONSE: *In the revised script (Fig.2 and Fig.1 4d & Page9), TEM and Raman data were supplemented.*

And we also add the corresponding discussion (Line 9-22 & Page 8).

Question 8: *Fig 4- the images show different magnifications. Authors should check their magnifications and scale bars. a and b look very zoomed out (low magnification) as compared to c-e.*

RESPONSE: *We are very sorry for our incorrect magnifications and scale bars. The incorrect picture has been deleted and the correct picture has been updated. (Now Fig.5 in the revised script)*

Question 9: Table 2 - It seems that GO-g-PEG works better than GC counterparts. First, include the reasoning in the manuscript about it and secondly, what is the point of doing carboxylation when there is a detrimental effect of it. Need to include proper explanation and discussion in the manuscript

RESPONSE: We are very sorry for our inadequate discussion.

Actually, we think GO-g-PEG is not better than GC-g-PEG as a nanofiller for PLA to some extent. Because at the same content (0.1%-0.3%), the Elongation at Break of PLA/GC-g-PEG is higher than PLA/GO-g-PEG. And we used 0.2wt% GC-g-PEG can reach 18%, which is equal to 0.3wt% GO-g-PEG, so we can use less nanofiller to get the same effect. Moreover, when the content of GC-g-PEG is 0.3%, the value can reach 31%, which is higher than all PLA/GO-g-PEG nanocomposites.

But when we increase the content of GC-g-PEG, the value decreased to 17%. All in all, we want to use less nanofiller to get better performance.

Samples	Tensile Strength(Mpa)	Elongation at Break(%)	Samples	Tensile Strength(Mpa)	Elongation at Break(%)
PLA	63	4	PLA	63	4
PLA/GC-g-PEG 0.1	61	13	PLA/GO-g-PEG 0.1	62	10
PLA/GC-g-PEG 0.2	60	18	PLA/GO-g-PEG 0.2	61	16
PLA/GC-g-PEG 0.3	59	31	PLA/GO-g-PEG 0.3	61	18
PLA/GC-g-PEG 0.4	58	17	PLA/GO-g-PEG 0.4	60	22
			PLA/GO-g-PEG 0.5	59	14

And we also add the corresponding discussion (Line 24-29 and Line 1-8 & Page 16-17).

Question 10: It would be interesting to see the mechanical properties of

PLA with just GO and GC without PEG.

RESPONSE: *Considering the Reviewer's suggestion, we have added PLA/GO and PLA/GC nanocomposites in Support Information. (Table S3 and Fig. S4) (Line 13-14 & Page 16 revised script)*

Question 11: *"7a and 7b that pure PLA exhibits a smooth and nondescript fracture surface, revealing a typical behavior of brittle fracture" - not very convincing from the SEM images presented. The features in 7a looks quite similar to fig 4a. It is difficult to accept it as a fracture, instead it seems like a film drying effect. The features identified as cracks are everywhere in the film it seems and in different directions which again indicate drying effect. Judging by the two images, 4a and 7a, the scale bar in 4a seems incorrect.*

RESPONSE: *We do understand the reviewer's concerns about the features of SEM images. We re-selected cryofracture SEM graphs. We hope our pictures will satisfy you.*

To be honest, the Fig. 4 (Fig.5 in revised script) is a cryofracture SEM graph, we immersed the solid samples rather than films into the liquid nitrogen, and then used two pliers to split it from the middle.

Similar phenomena can be seen in the study of xu et al, what they are studying is about crystallization (Fig s5)

(https://pubs.acs.org/doi/suppl/10.1021/acs.macromol.5b00462/suppl_file/ma5b00462_si_001.pdf)

We have also corrected the tensile section of pure PLA in Fig. 7a (Fig.10a in revised script). And we are very sorry for our mistakes about the scale bars. Now they are correct.

By the way, we are very sorry for the unclear expression of the experimental details. There is no drying step from liquid to solid in our nanocomposites preparation, only the preparation of nanofillers involves the evaporation of solvents. Moreover, both melt-blending and hot-pressing processes in our case were carried out at about 170~180°C that was by far higher than boiling temperature of Chloroform/ Ethanol solvent (around 61°C).

Also we apologize for the unclear expression, the nanocomposites prepared in our experiments are block solid. (as shown in the figure above). For clarity, we put the digital picture into the stress-strain curve. (Fig. 8 in revised script)

Question 12: *Fig 7, how much stress was applied? Include more low magnification images to show such features are limited to the fracture site. Also include images under different stress conditions to show the increase in the 'said' voids/crack sites. These images are not convincing. It appears the features are intrinsically present in the films.*

RESPONSE: *The stress we used is slowly increasing, from 0 to the maximum of 10KN, and in our experiments, the maximum force is about 1.2KN, the tensile rate was 5 mm min⁻¹.*

As for seeing more fractures at different locations at lower magnifications, due to this COVID-19, neither our samples nor the experimental machine can be used, nor can we go back to school. And this article is very urgently needed too, we would like to provide the experimental data, but we are limited by the force majeure, so we are very sorry that we really have no way to provide the SEM image of nanocomposites in different locations.

The following two pictures are the low magnification SEM pictures of PLA(a) and PLA/GC-g-PEG 0.3 nanocomposite(b). We can found that pure PLA is a relatively smooth section. After adding GC-g-PEG, many holes appear on the surface.

We also added these picture in the supplementary information (Fig. S5 in Supplementary Material).

Then one of the most obvious features that can indicate the improvement of ductility is that the middle of samples will turn white, as shown in the picture below.

And because the voids and cracks that are on the fracture section, if the force is too small, the sample cannot break down, and it is difficult to see the fracture surface. So, we are waiting for PLA to break down on the stretching machine, then cut out a cross-section to do the SEM.

Question 13: *Fig 8, there is no evidence shown to iterate such scheme. Where are PEG-GC/GO are worm like structures or any morphology for that matter. Why would the crack initiate at the PEG/GC/GO? Wouldn't it initiate in PLA layer which has higher Tg and then preferentially propagate through polymer layer itself. There is only 0.3% of filler here. This schematic and explanation do not support the experimental data.*

RESPONSE: *We are very sorry for our incorrect writing this scheme, and now we have removed it.*

And thank you again for pointing this problem out.

Question 14: *Fig 9, this marginal reduction in T5 cannot be regarded as heat dissipation mechanism. Any filled would do the job. Recommended to include the data for just GO and GC without PEG, to show that PEG is vital for any of these phenomena described in this study. SEM images are not convincing enough to highlight the homogeneous distribution of functionalised GC/GO sheets within PLA. Can authors include any convincing evidence such as cross-section TEM?*

RESPONSE: *Considering the Reviewer's suggestion, we have added PLA/GC 0.3 without PEG for comparison, and we can see that without*

PEG, the PLA/GC 0.3 nanocomposite is very similar to pure PLA.(Line 28-29 &Page20 and Fig.11 & Page21 in revised script)

And we also did the TEM characterization of PLA/GC-g-PEG 0.3 (Fig.6 in Page 13). And we also add the corresponding discussion (Line 9-12 &Page12)

<Reviewer #2>

Comments: *The authors described a process of carboxylation and content on PLA as well as the comparison of GO-g-PEG; the new materials were characterized by a variety of means. New PLA nanocomposites were prepared by using GC-g-PEG as a nanofiller. The interaction between PEG and PLA led to better dispersion. Crystallization behavior shows that the addition of nanofillers can accelerate the cold crystallization process and improve the crystallinity of PLA. The addition of GC-g-PEG at 0.3%, the elongation at break of PLA increased to 30.55%. The good compatibility between GC-g-PEG and PLA and the uniform dispersion in PLA matrix are mainly attributed to the improvement of the ductility of PLA while maintaining the tensile strength. Overall I found the work suitably significant and comprehensive for publication. I do have a few queries the authors should attend to:*

Question 1. *In a number of places the English should be corrected for improved reading.*

RESPONSE: *We are very sorry for our inappropriate statement. Referee #1 also proposed some improvements to some parts of our manuscript. We also carefully checked the related statements. Thank you for your correction.*

Question 2: *The authors claim PLA is a suitable alternate as a biobased polymer, what about cellulose, which is the most abundant on earth?*

RESPONSE: *Judging from our current knowledge, we think the cellhouse is the most abundant on earth, because cellulose is an inexhaustible natural polymer material synthesized mainly by photosynthesis in plants in nature. We think both PLA and cellulose will be main materials for future environmental protection.*

Question 3: *What is meant by high-intensity in the context of mechanical testing?*

RESPONSE: *We are so sorry for this wrong expression. What we wanted to express at the beginning was high strength. Reviewer #1 also mentioned this issue, and we have corrected this part.*

In the revised script (Line 21-23 & Page 2) , we have re-written this part according to the Reviewer's suggestion.

Unfortunately, as to PLA, because of its intrinsic brittleness and low tensile elongation at break, the application of PLA is still limited in spite of its high elastic modulus and tensile strength

Question 4: *Mpa should be MPa throughout*

RESPONSE: *We are very sorry for our incorrect writing Mpa, Now we have replaced all Mpa with MPa.*

Question 5: *hakke rheomix should be capitalized and the country of all equipment should be stated.*

RESPONSE: *Considering the Reviewer's suggestion, hakke rheomix has been capitalized. (Line18 & Page 5)*

Also we have added the country of all equipment.(Line 18, 27, 29 & Page 5 and Line 3, 5, 7, 12, 22, 25, 26 & Page 6)

<Reviewer #3>

Comments: The manuscript reports on the improvement of the mechanical properties of PLA by the chemical functionalization of GO with carboxylic and PEG groups. The manuscript is well written, the experiments are well performed and the conclusions are supported by experimental results. I suggest the publication of the present work in RSOS as its present form.

RESPONSE: *Thank you very much for your recognition of our work.*